# *Silene uniflora* Extracts for Strawberry Postharvest Protection

**DOI:** 10.3390/plants12091846

**Published:** 2023-04-29

**Authors:** Laura Buzón-Durán, Eva Sánchez-Hernández, Pablo Martín-Ramos, Luis Manuel Navas-Gracia, Mari Cruz García-González, Rui Oliveira, Jesús Martín-Gil

**Affiliations:** 1Department of Agricultural and Forestry Engineering, ETSIIAA, Universidad de Valladolid, 34004 Palencia, Spain; laura.buzon@uva.es (L.B.-D.); pmr@uva.es (P.M.-R.); luismanuel.navas@uva.es (L.M.N.-G.); jesus.martin.gil@uva.es (J.M.-G.); 2Department of Agroforestry Sciences, ETSIIAA, Universidad de Valladolid, Avenida de Madrid 44, 34004 Palencia, Spain; mariacruz.garcia.gonzalez@uva.es; 3Centre of Molecular and Environmental Biology (CBMA), Department of Biology, School of Sciences, University of Minho, Campus de Gualtar, 4710-057 Braga, Portugal; ruipso@bio.uminho.pt

**Keywords:** antifungal, anthracnose, chitosan oligomers, grey mold, halophyte, sea campion

## Abstract

Halophytes are gaining considerable attention due to their applications in saline agriculture, phytoremediation, medicine, and secondary metabolite production. This study investigated the bioactive components present in *Silene uniflora* (sea campion) hydromethanolic extract, and their antimicrobial activity was evaluated both in vitro and ex situ against two strawberry phytopathogens, namely *Botrytis cinerea* (grey mold) and *Colletotrichum nymphaeae* (anthracnose fruit rot). The main identified phytochemicals were mome inositol, saturated fatty acid esters, and cyclotetracosane. In vitro tests demonstrated complete inhibition of the growth of *B. cinerea* and *C. nymphaeae* at extract concentrations of 1000 and 1500 μg·mL^−1^, respectively, with an activity comparable to that of fosetyl-Al and substantially higher than that of azoxystrobin. This activity was improved upon conjugation with chitosan oligomers (COS), yielding inhibition values of 750 and 1000 μg·mL^−1^. The COS-*S. uniflora* conjugate complexes were then tested as protective treatments for postharvest storage of strawberry fruit, resulting in high protection against artificially inoculated *B. cinerea* and *C. nymphaeae* at doses of 3750 and 5000 μg·mL^−1^, respectively. The reported results open the door to the valorization of this halophyte as a source of biorationals for strawberry protection.

## 1. Introduction

*Silene uniflora* Roth is an herbaceous perennial plant of the Caryophyllaceae family, typically forming mats on cliffs. Its leaves are linear, gray-green, and glaucous, and the flowers are white with five sepals that form a bladder and five deeply notched petals (Figure 1a).

Limited research has been conducted on the chemical composition of *Silene* spp. Triterpene saponins and pectic polysaccharides have been isolated from *Silene vulgaris* (Moench) Garcke [1,2,3], and the oil compositions of various *Silene* species have also been investigated [4,5,6]. The phytochemical profiling of various extracts from species in the *Silene* genus has demonstrated the presence of sterols, alkaloids, tannins, flavonoids [7,8], and phytoecdysteroids (such as 2,22-dideoxyecdysone 25-*O*-*β*-d-glucopyranoside, Figure 1b) [9,10,11]. Coumaric acid derivatives and catechin were reported in the leaves and procyanidin B1 in the root extract of *S. vulgaris* subsp. *macrocarpa* [12]. Quinic, malic, protocatechuic, and *p*-coumaric acids, as well as hesperidin, were identified in six *Silene* species (i.e., *Silene alba* (Mill.) E.H.L.Krause, *Silene conoidea* L., *Silene dichotoma* Ehrh., *Silene italica* (L.) Pers., *Silene supina* M.Bieb., and *S. vulgaris*) [13]. However, the phytochemical profile of *S. uniflora* has yet to be reported.

Regarding antimicrobial activity, there have been promising reports on *Silene parishii* S.Watson [14], *S. vulgaris* [15,16], *Silene cariensis* Boiss. [17], *S. alba*, *S. conoidea*, *S. dichotoma*, *S. italica*, *S. supina*, and *S. vulgaris* [13], but there is a dearth of information on the activity of *S. uniflora*.

The aim of the study presented herein was two-fold: (i) to investigate the phytoconstituents of *S. uniflora* using vibrational spectroscopy (IR) and gas chromatography–mass spectrometry (GC-MS) and (ii) to evaluate *S. uniflora* antifungal activity, alone and in combination with chitosan oligomers (COS), and open new pathways for its valorization. In particular, two of the most important strawberry (*Fragaria* × *ananassa* Duch.) pathogens were selected: *Botrytis cinerea* Pers. and *Colletotrichum nymphaeae* (Pass.) Aa., which ranked second and eighth on a list of fungal pathogens of scientific and economic importance, respectively [18]. The former (grey mold) has a wide host range (over 200 plant species) and potential for causing severe damage, both pre- and post-harvest; the latter belongs to the *Colletotrichum acutatum* J.H.Simmonds clade and causes anthracnose [19]. The reported findings may contribute to the management of these diseases in agricultural ecosystems.

## 2. Results

### 2.1. Vibrational Characterization

Prior to extract preparation, infrared spectra of the different aerial plant organs of *S. uniflora* (Appendix A) were analyzed in an initial screening to identify functional groups and assess the presence/absence of significant differences among them. The fingerprint regions of the three spectra were very similar. Band assignments are summarized in Table 1. The band at 3366 cm^−1^ can be attributed to hydrogen bonding in pyranosides [20], and those at ca. 2916, 1443, 1371, 1244, and 1146 cm^−1^ are consistent with the presence of inositol, detected by GC-MS, as discussed below. Concerning the presence of saturated fatty acid vinyl esters, this is supported by the bands at 2916, 2848, 1636, 1472, 1417, 1378, 1307, 1243, 1146, 1101, and 719 cm^−1^.

### 2.2. GC-MS Characterization

The main phytochemicals identified in the extract prepared from a mixture of the aerial organs of the plant (Table 2, Appendix A) were: 4-*O*-methyl-*myo*-inositol or mome inositol (52.5%), saturated fatty acid vinyl esters (8.7%, Appendix A), and cyclotetracosane (3.7%), depicted in Figure 2.

### 2.3. In Vitro Growth Inhibition Tests

The results of the mycelial growth inhibition tests (Appendix A) are summarized in Figure 3. When tested separately, *S. uniflora* showed greater efficacy against *B. cinerea* than COS, as full inhibition was reached at 1000 and 1500 μg·mL^−1^, respectively. For *C. nymphaeae*, both *S. uniflora* and COS exhibited approximately the same efficacy (MIC = 1500 μg·mL^−1^). An enhancement in terms of efficacy was observed in both cases for the COS-*S. uniflora* extract conjugate complex, reaching full inhibition at 750 and 1000 μg·mL^−1^ for *B. cinerea* and C. *nymphaeae*, respectively.

Upon comparison of the effective concentrations (Table 3), it was possible to observe differences in the efficacy of the treatments more clearly. The highest efficacy (i.e., the lowest EC_50_ and EC_90_ values) was observed for the COS-*S. uniflora* conjugate complex against *B. cinerea*, followed by those of the same treatment against *C. nymphaeae.* Synergistic behavior was found between COS and *S. uniflora* extract (SF values ≥ 1), with the highest synergy factor being obtained for the EC_90_ of COS-*S. uniflora* against *B. cinerea* (SF = 1.56).

Table 4 summarizes the inhibition results for three synthetic fungicides. Mancozeb completely inhibited the growth of both fungal pathogens even at a tenth of the recommended dose (i.e., at 150 μg·mL^−1^), while Fosetyl-Al required a concentration of 2000 μg·mL^−1^, and azoxystrobin did not fully inhibit the growth of the two fungal taxa at 62,500 μg·mL^−1^.

### 2.4. Ex Situ Growth Inhibition Tests

Strawberry fruits of the variety “Calinda” were treated with the most active product according to the in vitro assays, the COS-*S. uniflora* conjugate complex, at MIC×5 (namely 3750 and 5000 μg·mL^−1^ for *B. cinerea* and *C. nymphaeae*, respectively). As depicted in Figure 4, the treatment noticeably reduced the incidence of both pathogens. Disease incidences were calculated on days 1, 7, and 10 of the experiment (Table 5). In the negative controls, the pathogens did not proliferate (thus ruling out the possibility of contamination), whereas in the strawberries that had been artificially inoculated with the pathogens but not treated (positive controls), *B. cinerea* and *C. nymphaeae* were able to invade more than 81% of the surface of all fruits on the tenth day, with an incidence of 5 at the end of the trial. Upon treatment with the COS-*S. uniflora* extract conjugate complex, an incidence of 1.3 was observed on the tenth day, with the most-affected fruits showing a colonization of less than 40% by *B. cinerea*, whereas the colonization of fruits artificially infected with *C. nymphaeae* was higher, with an incidence of 2.3 on the tenth day (i.e., most fruits showed a colonization equal to or higher than 40%).

Concerning fruit quality attributes, the COS-*S. uniflora* extract treatment exerted a beneficial effect on the firmness, with an average 24% decrease in flesh firmness values in the case of *B. cinerea* and a 33% decrease for *C. nymphaeae* vs. a 52% decrease in the untreated fruits (negative control) by the end of the experiment. As far as color is concerned, the COS-*S. uniflora* coating imparted a slightly paler shade of red on day 10, more evident in the fruits inoculated with *B. cinerea* than in those inoculated with *C. nymphaeae*, although quantitative color measurements would be needed to determine the actual impact on the hue degree and chroma. This should be taken into account, as it may influence consumer preferences.

## 3. Discussion

### 3.1. On the Phytochemical Profile

Given that only a small subset of the known organic compounds (amenable for GC-MS) is present in the largest mass spectral databases, limitations in the identification of some of the compounds present in the extracts were detected, with quality of resemblance (Qual) values below 80 (Table 2). Caution is advised as identification of such compounds may be unreliable. However, it is noteworthy that for the chemical species identified at RT = 26.8367 min, comprising 8.7% of the peak area with a “Qual” value of 41, its MS spectrum shows good agreement with those of myristic acid vinyl ester and palmitic acid vinyl ester (Appendix A), and the bands identified in the FTIR spectra of the dried aerial parts also support the presence of saturated fatty acid vinyl esters.

Concerning the presence of the main identified compounds in other plant extracts, mome inositol has previously been reported in high amounts in *Corbichonia decumbens* (Forssk.) Exell (49.5–75.5% depending on the plant organ) [21], *Clitoria ternatea* L. (38.7%) [22], *Spergula arvensis* L. (38.1%) [23], *Nephelium lappaceum* L. (36%) [24], and *Macrotyloma uniflorum* (Lam.) Verdc. (23.2%) [25]. This phytoconstituent is anti-alopecic, anti-cirrhotic, anti-neuropathic, cholesterolytic, lipotropic, and a sweetener [25].

With regard to vinyl palmitate and vinyl myristate, they have been documented in *Simarouba glauca* DC. [26], *Eichhornia crassipes* (Mart.) Solms [27], Phymatosorus scolopendria (Burm.fil.) Pic.Serm. [28], *Cinnamomum javanicum* Blume [29], and *Petiveria alliaceae* L. extracts [30].

In comparison to other salt-tolerant plants (namely *Crithmum maritimum* L. [31], *Daucus carota* subsp. *gummifer* (Syme) Hook. fil. [31], *Tripleurospermum callosum* (Boiss. and Heldr.) E.Hossain [32], *Limonium binervosum* (G.E.Sm.) C.E.Salmon [33] and *Tamarix gallica* L. [34]), it can be noted that the extract of a mixture of the aerial parts of *S. uniflora* shares with *T. callosum* and *L. binervosum* the presence of cyclotetracosane at a moderate concentration (3%). In those plant extracts in which cyclotetracosane is present in quantities greater than 10%, such as the essential oil of *Valeriana officinalis* L., the ethanol and methanol extracts of *Cyclosorus dentatus* (Forssk.) Ching [35] (an allelopathic plant), or the ethyl acetate root extract of *Jatropha zeyheri* Sond. [36], substantial antioxidant capacities have been reported. Cyclotetracosane has been demonstrated to possess *α*-amylase inhibitory activity [37,38].

### 3.2. On the Antimicrobial Activity

#### 3.2.1. Activity of Other *Silene* spp. Extracts

In terms of activity against the two phytopathogens studied here, data are only available for methanolic *Silene armeria* L. leaf extract [39], for which a MIC value of 1000 μg·mL^−1^ was reported against *B. cinerea* and *Colletotrichum capsici* (Syd. and P.Syd.) E.J.Butler and Bisby, comparable to the MICs reported herein (1000 and 1500 μg·mL^−1^ against *B. cinerea* and *C. nymphaeae*, respectively). This same *S. armeria* extract achieved MIC values in the 500–2000 μg·mL^−1^ range against *Rhizoctonia solani* Kühn, *Fusarium oxysporum* Schlechtendal, *Fusarium solani* W.C.Snyder, *Sclerotinia sclerotiorum* (Lib.) de Bary, and *Phytophthora capsici* Leonian, suggesting that similar activity may be expected for the *S. uniflora* extract studied here.

Concerning other *Silene* spp. extracts, there are reports on their antimicrobial activity against other microorganisms. Back in 1993, Hoffmann et al. [14] showed that the ethanolic extract of *S. parishii* at a concentration of 1000 μg·mL^−1^ was effective against *Bacillus subtilis* (Ehrenberg, 1835) Cohn, 1872, partially effective against *Candida albicans* (C.P.Robin) Berkhout, and had no effect on *Staphylococcus aureus* Rosenbach, 1884 and *Klebsiella pneumoniae* (Schroeter, 1886) Trevisan, 1887. Subsequent studies by Boukhira et al. [15] and Thakur et al. [16] on *S. vulgaris*, Keskin et al. [17] on *S. cariensis* subsp. *cariensis* and *S. pungens*, and Zengin et al. [13] on *S. alba*, *S. conoidea*, *S. dichotoma*, *S. italica*, *S. supina*, and *S. vulgaris* have demonstrated that *Silene* spp. extracts have significant antibacterial and antifungal activities against *S. aureus*; *Bacillus cereus* Frankland and Frankland, 1887; *Escherichia coli* (Migula, 1895) Castellani and Chalmers, 1919; *Pseudomonas aeruginosa* (Schroeter, 1872) Migula, 1900; *Aeromonas hydrophila* (Chester, 1901) Stanier, 1943; *Salmonella enterica enterica* (ex Kauffmann and Edwards, 1952) Le Minor and Popoff, 1987; *Listeria monocytogenes* (Murray et al., 1926) Pirie, 1940; *Enterococcus faecalis* (Andrewes and Horder, 1906) Schleifer and Kilpper-Bälz, 1984; *Micrococcus flavus* Liu et al., 2007; *C. albicans*; *Aspergillus brasiliensis* Varga et al.; *Aspergillus versicolor* (Vuillemin) Tiraboschi; *Aspergillus fumigatus* Fresenius; *Aspergillus ochraceus* K.Wilhelm; *Aspergillus niger* van Tieghem; *Penicillium ochrochloron* Biourge; *Penicillium funiculosum* Thom; *Penicillium verrucosum* Dierckx; and *Trichoderma viride* Persoon.

#### 3.2.2. Comparison with Synthetic Antimicrobials

The concentrations of *S. uniflora* extract required for full inhibition of *B. cinerea* and *C. nymphaeae* (1000 and 1500 μg·mL^−1^, respectively; see Table 3) were an order of magnitude higher than those of mancozeb (Table 4), demonstrating a substantially lower antimicrobial activity. The activity of *S. uniflora* extract was comparable to that of fosetyl-Al (MIC = 2000 μg·mL^−1^), but substantially higher than that of azoxystrobin (MIC > 62,500 μg·mL^−1^).

#### 3.2.3. Comparison with Chitosan-Based Coatings for Postharvest Strawberry Protection

In order to compare the protective effect of the COS-*S. uniflora* extract conjugate complex with other chitosan-based coatings reported in the literature (on strawberries), the results of a brief bibliographical survey are presented in Table 6, itemized into those used for postharvest control of gray mold decay (*B. cinerea*) and those aimed at anthracnose (*Colletotrichum* spp.) control. In the case of *B. cinerea*, it may be observed that the efficacy of the COS-*S. uniflora* extract treatment (3750 and 5000 μg·mL^−1^ against *B. cinerea* and *C. nymphaeae*, respectively) was markedly superior to those reported for chitosan acetate, chitosan chloride, chitosan glutamate, and chitosan formate, in which a 1% *w*/*v* dose was applied, with higher disease severities at the end of the experiments [40]. However, it was not as effective as *Zataria multiflora* Boiss. essential oil encapsulated in chitosan nanoparticles (1500 µg·mL^−1^) [41] and COS-*Uncaria tomentosa* (Willd. ex Schult.) DC conjugate complexes (1000 µg·mL^−1^).

When it comes to protection against *C. nymphaeae*, to the best of our knowledge, no previous studies on postharvest protection using chitosan have been reported for strawberry fruits. Comparison with other treatments against *Colletotrichum* spp. reported in the literature reveals the efficacy of COS-*S. uniflora* would be similar to those reported by Arceo Martínez et al. [42] for chitosan at a concentration of 7500 µg·mL^−1^.

**Table 6 plants-12-01846-t006:** Summary of chitosan-based treatments used for postharvest control of gray mold (*B. cinerea*) and anthracnose (*Colletotrichum* spp.) on strawberry fruits reported in the literature and their associated disease severities.

Pathogen		Chitosan Complex	Storage Conditions	Disease Severity (0–5)	Ref.
*B. cinerea*		Chitosan + *Silene uniflora* (3750 μg·mL^−1^)	7 days at 4 °C, followed by 3 days at 20 °C	1.3	This work
		Chitosan acetate (1% *w*/*v*)	4 days at 20 ± 1 °C, 95–98% RH	3.1	[40]
		Chitosan chloride (1% *w*/*v*)		3.2	
		Chitosan formate (1% *w*/*v*)		3.4	
		Chitosan glutamate (1% *w*/*v*)		3.4	
		Commercial chitosan (1% *w*/*v*)		3.5	
		Chitosan (1% *w*/*v*)	7 days at 0 ± 1 °C, 95–98% RH, followed by 3 days of shelf life at 20 ± 1 °C, 95–98% RH	2.7	
		Chitosan NP (1500 μg·mL^−1^)	7 days at 4 °C, followed by 2 days at 20 °C	2.6	[41]
		Chitosan NP + *Zataria multiflora* (1500 μg·mL^−1^)		1.5	
		Chitosan + *Cinnamomum zeylanicum* (1500 μg·mL^−1^)		2.4	[43]
		Chitosan + *Z. multiflora* (1500 μg·mL^−1^)		1.5	
		COS + *Uncaria tomentosa* (100 μg·mL^−1^)	7 days at 4 °C, followed by 3 days at 20 °C	3.5	[44]
		COS + *U. tomentosa* (500 μg·mL^−1^)		1.7	
		COS + *U. tomentosa* (1000 μg·mL^−1^)		0.5	
*Colletotrichum* spp.	*C. nymphaeae*	Chitosan + *S. uniflora* (5000 μg·mL^−1^)	7 days at 4 °C, followed by 3 days at 20 °C	2.3	This work
	*C. gloeosporioides*	Chitosan (7500 μg·mL^−1^)	7 days at 2 ± 2.0 °C, followed by 3 days at 25 ± 2.0 °C	2	[42]
		Chitosan (10,000 μg·mL^−1^)		1.2	
		Chitosan (15,000 μg·mL^−1^)		1	
	*C. acutatum*	Chitosan (7500 μg·mL^−1^)	7 days at 2 ± 2.0 °C, followed by 3 days at 25 ± 2.0 °C	2	
		Chitosan (10,000 μg·mL^−1^)		1.8	
		Chitosan (15,000 μg·mL^−1^)		1	
	*C. fragariae*	Chitosan + cinnamon EO + aqueous extract of *Roselle calyces*	Stored at two different temperatures (5 and 20 °C) for 10 d	1 at 5 °C	[45]
				5 at 20 °C	

#### 3.2.4. Mechanism of Action

Based on the activities referred to in the literature for the main constituents identified in *S. uniflora* extract, the observed antifungal activity should be mainly ascribed to the presence of 4-*O*-methyl-*myo*-inositol. Although there are no reports on the antimicrobial activity of pure 4-*O*-methyl-*myo*-inositol, the aforementioned extract of *N. lappaceum* showed antibacterial activity against food pathogenic and spoilage bacteria [24]. Furthermore, *myo*-inositol has demonstrated strong antifungal activity against *Fusarium circinatum* Nirenberg and O’Donnell, *Cryphonectria parasitica* (Murril) M.E. Barr, and *Phytophthora cinnamomi* Rands phytopathogens, with MIC values of 1000, 750, and 375 μg·mL^−1^, respectively [46]. Both vinyl palmitate and vinyl myristate have also shown antimicrobial activity [27,47]. However, the contributions from other minority constituents and synergistic behaviors among them cannot be ruled out.

Concerning COS, its antifungal activity is well-established [48] and is thought to be due to its positive charge interacting with the negative charge of the fungal cell membrane. This interaction leads to increased permeability of the cell [49], resulting in a loss of intracellular components that disrupts the osmotic pressure and causes cell death [50]. COS can also alter chitin levels, leading to a weakened cell wall [51], and can generate ROS that damage biomolecules, triggering apoptosis and necrosis. Additionally, COS can interfere with DNA and RNA synthesis [52].

With regard to enhanced activity upon the formation of conjugate complexes, the observed synergism may stem from an enhanced additive fungicidal activity per se or by simultaneous action at multiple fungal metabolic sites [53], but it may also be due to the fact that chitosan oligomers can increase the solubility and bioavailability of the bioactive compounds present in the extract.

## 4. Materials and Methods

### 4.1. Plant Material and Chemicals

Samples of *S. uniflora* were collected in May 2021, during the flowering stage (Figure 1a), from Playa de Cué (Llanes, Asturias, Spain; 43°24′58.7″ N 4°43′53.3″ W). Specimens were identified and authenticated by Prof. Dr. Baudilio Herrero Villacorta (Departamento de Ciencias Agroforestales, ETSIIAA, Universidad de Valladolid) and voucher specimens are available from the herbarium of the ETSIIAA. Aerial parts from different specimens (*n* = 20) were mixed to obtain (separate) representative composite samples of flowers, fruits, and leaves. The composite samples were shade-dried (with a 72% weight loss), reduce to powder using a mechanical grinder, homogenized, and sieved (1 mm mesh).

Strawberry fruits (*Fragaria* × *ananassa* cv. “Calinda”) were supplied by Ideal Fruits (Chañe, Segovia, Spain). The fruits were produced without the addition of artificial pesticides in accordance with organic farming regulations. The fruits were collected and transported to the laboratory in refrigerated conditions, and ex situ tests began within 24 h of harvesting. Strawberries were chosen on the bases of uniform size, lack of physical damage and fungal infection, and a red coloration covering more than 75% of the surface, in agreement with Romanazzi et al. [40].

Potato dextrose agar (PDA) was provided by Becton Dickinson (Bergen County, NJ, USA), Neutrase^TM^ 0.8 L enzyme was acquired from Novozymes A/S (Bagsværd, Denmark), and high-molecular-weight chitosan (CAS 9012-76-4) was purchased from Hangzhou Simit Chem. and Tech. Co. (Hangzhou, China).

For comparison purposes, the Plant Health and Certification Service of the Government of Aragon provided commercial fungicides, namely Ortiva^®^ (azoxystrobin 25%; Syngenta), Vondozeb^®^ (mancozeb 75%; UPL Iberia), and Fesil^®^ (fosetyl-Al 80%; Bayer).

### 4.2. Fungal Isolates

The fungal isolates of *B. cinerea* (code not available, but details on its provenance are provided in [54]) and *Colletotrichum nymphaeae* were supplied as subcultures in PDA by Richerd Breia and Hernâni Gerós from the Centre of Molecular and Environmental Biology (CBMA) at the University of Minho and by Pedro Talhinhas, School of Agriculture, University of Lisbon, respectively. They were cultured in PDA at 25 °C in the dark.

### 4.3. Preparation of S. uniflora Extract, Chitosan Oligomers, and Their Conjugate Complex

The extract preparation procedure was similar to the one previously reported in [31]. Briefly, 20 g of dried *S. uniflora* flowering aerial parts were mixed with a 300 mL methanol/water solution (1:1 *v*/*v*) and heated in a water bath at 50 °C for 30 min. Then, the solution was subjected to sonication for 5 min in pulse mode with a 1 min stop every 2.5 min, using a model UIP1000hdT probe-type ultrasonicator (Hielscher Ultrasonics; Teltow, Germany). The solution was centrifuged at 9000 rpm for 15 min and the supernatant was filtered through Whatman No. 1 paper, followed by freeze-drying to obtain the solid residue. The extraction yield was 5%. For subsequent GC-MS analysis, 25 mg of the obtained freeze-dried extracts were dissolved in 5 mL of HPLC-grade MeOH to obtain a 5 mg·mL^−1^ solution, which was further filtered.

Chitosan oligomers (COS) were prepared using the method described in [55] with the modifications described in [56], yielding oligomers of molecular weight < 2000 Da in a solution with a pH of 4.5. COS and *S. uniflora* extract solutions (150 mL of each solution, both at a concentration of 3000 μg·mL^−1^) were mixed in a 1:1 (*v*/*v*) ratio and sonicated for 15 min in five 3 min pulses to obtain the conjugate complexes.

### 4.4. Characterization Procedures

The infrared spectra of the *S. uniflora* dried plant organs were registered using a model Nicolet iS50 Fourier transform infrared spectrometer from Thermo Scientific (Waltham, MA, USA), equipped with a diamond attenuated total reflection (ATR) system. The spectra were acquired at 1 cm^−1^ spectral resolution over the 400–4000 cm^−1^ range by co-adding 64 scans.

The hydromethanolic extract was studied using GC-MS at the Research Support Services (STI) at Universidad de Alicante (Alicante, Spain), utilizing a model 7890A gas chromatograph coupled to a model 5975C quadrupole mass spectrometer (Agilent Technologies, Santa Clara, CA, USA). The chromatographic conditions consisted of an injection volume of 1 µL; an injector temperature of 280 °C, in splitless mode; and an initial oven temperature of 60 °C for two minutes, followed by a 10 °C per minute ramp up to a final temperature of 300 °C, kept for 15 min. The chromatographic column used for the separation of the compounds was an HP-5MS UI of 30 m length, 0.250 mm diameter, and 0.25 µm film. The mass spectrometer conditions were set to a temperature of the electron impact source of 230 °C, and the quadrupole was set to 150 °C; the ionization energy was set to 70 eV. The identification of phytoconstituents was based on a comparison of their mass spectra and retention time with those of authentic compounds and by computer matching with the National Institute of Standards and Technology (NIST11) and Wiley databases.

### 4.5. Antifungal Activity Assessment

#### 4.5.1. In Vitro Tests of Mycelial Growth Inhibition

The antimicrobial activities were evaluated using the agar dilution method [57]. Petri dishes with PDA incorporating ten different concentrations (ranging from 62.5 to 1500 μg·mL^−1^) of the various treatments—namely COS, *S. uniflora* extract, and COS-*S. uniflora* extract—were inoculated with 5 mm plugs and cultured at 25 °C for seven days. The control consisted in replacing the extract with the solvent used for extraction (i.e., methanol:water 1:1 *v*/*v*) in the PDA medium. Tests with commercial fungicides were performed in parallel and using the same source of inoculum. Growth inhibition was calculated by the following formula: ((*d_c_* − *d_t_*)/*d_c_*) × 100, where *d_c_* is the average colony diameter in the control colony and *d_t_* is the average colony diameter in the treated colony. The 50% and 90% effective concentrations, EC_50_ and EC_90_, were calculated using IBM (Armonk, NY, USA) SPSS Statistics v.25’s PROBIT analysis. The synergy factor [58], which measures the degree of interaction, was estimated using Wadley’s method [59].

#### 4.5.2. Ex Situ Tests of Mycelial Growth Inhibition

Under controlled laboratory conditions, the efficacy of the COS-*S. uniflora* extract conjugate complex was evaluated on artificially inoculated strawberry fruits. The protocol was slightly modified from that proposed by Sánchez-Hernández et al. [44]. The strawberries were disinfected for 2 min with a NaOCl 3% solution, then washed three times with sterile distilled water and dried in a laminar flow hood on sterile absorbent paper. The strawberries were divided into three homogeneous groups of 45 fruits (three repetitions with 15 fruits per repetition, treatment, and pathogen), with all fruits measuring more than 22 mm in diameter. One group was treated with COS-*S. uniflora* extract conjugate complex (at a concentration of MIC×5, i.e., at 3750 μg·mL^−1^ for *B. cinerea* or 5000 μg·mL^−1^ for *C. nymphaeae*), while the other groups served as the negative (no treatment and no pathogen) and positive (pathogen and no treatment) controls. Superficial wounds (ø = 5 mm) were made in the equatorial zone of each fruit, and the strawberries were then immersed in the COS-*S. uniflora* conjugate complex treatment for five minutes and dried at room temperature in a laminar flow hood, using sterile absorbent paper. In the superficial wounds, a plug of a PDA culture from *B. cinerea* or *C. nymphaeae* was placed (with the mycelium facing the fruit wound). Following Hernández-Muñoz et al. [60], the fruits were placed in covered plastic boxes and stored for seven days at 4 °C and 95–98% RH, then exposed to a 3-day shelf life at 20 °C and 95–98% RH. In accordance with Romanazzi et al. [40], the percentage of rotten strawberries and the disease severity (according to an empirical scale with six degrees: 0, healthy fruit; 1, 1–20% of fruit surface infected; 2, 21–40% of fruit surface infected; 3, 41–60% of fruit surface infected; 4, 61–80% of fruit surface infected; 5, more than 81% of surface infected, with sporulation) were recorded during storage.

Concerning quality attributes, firmness was measured in the central zone of the strawberries (previously sliced into halves) using a TA-XT2 Texture Analyzer (Stable Micro Systems, Godalming, UK) with a 5 mm diameter flat probe. The penetration depth was 5 mm and the cross-head speed was 5 mm·s^−1^.

### 4.6. Statistical Analyses

In vitro and ex situ results were analyzed via one-way analysis of variance (ANOVA) followed by Tukey’s post hoc test in IBM SPSS Statistics v.25.

## 5. Conclusions

The hydromethanolic extract of *S. uniflora* aerial parts (leaves, flower petals, and fruits) has 4-*O*-methyl-*myo*-inositol (52.5%), saturated fatty acid esters (8.7%), and cyclotetracosane (3.7%) as its main phytoconstituents, according to our GC-MS results. Upon testing of its in vitro activity against *B. cinerea* and *C. nymphaeae* strawberry pathogens, MIC values of 1000 and 1500 μg·mL^−1^ were obtained, which are comparable to those of the synthetic fungicide fosetyl-Al. A synergistic effect was observed upon conjugation of chitosan oligomers with the halophyte extract, resulting in MIC values of 750 and 1000 μg·mL^−1^. Concerning the use of the COS-*S. uniflora* conjugate complex in postharvest protection of strawberry fruits, a dose five times higher than the in vitro MIC was required to achieve high inhibition against the two phytopathogens after 10 days (with disease severities of 1.3 and 2.3 out of 5 for *B. cinerea* and *C. nymphaeae*, compared to 5 out of 5 for the non-treated fruits). This activity is one of the highest reported for chitosan-based coatings, suggesting that *S. uniflora* extract may be a suitable biorational for the protection of this crop.

## Figures and Tables

**Figure 1 plants-12-01846-f001:**
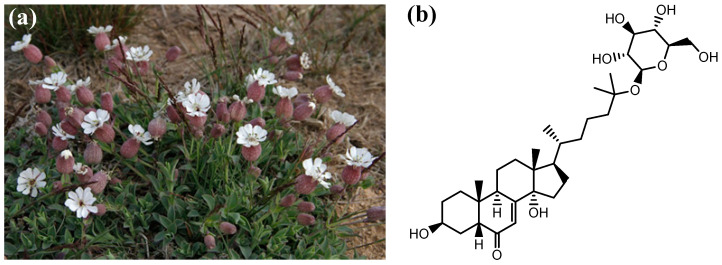
(**a**) Photograph of *Silene uniflora* during its flowering stage growing on a cliff in Playa de Cué (Llanes, Asturias, Spain); (**b**) chemical structure of 2,22-dideoxyecdysone 25-*O*-*β*-d-glucopyranoside phytoecdysteroid reported in *Silene* spp. extracts.

**Figure 2 plants-12-01846-f002:**
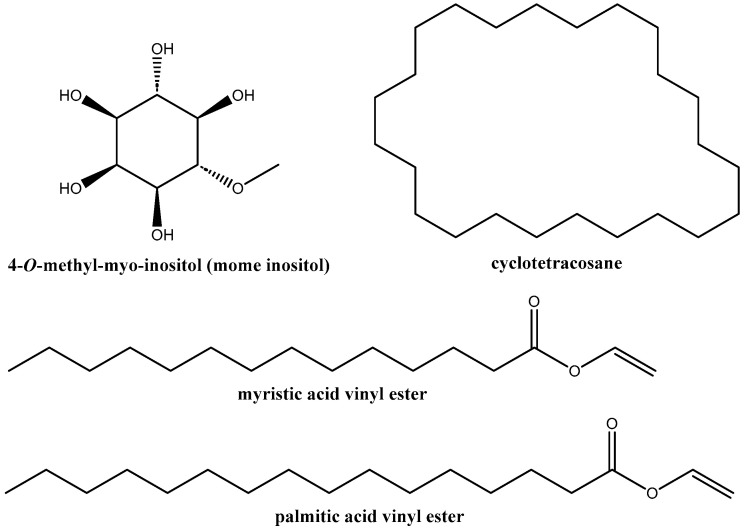
Chemical structures of the four most abundant phytochemicals identified using gas chromatography–mass spectrometry in the hydromethanolic extract of *S. uniflora* aerial parts.

**Figure 3 plants-12-01846-f003:**
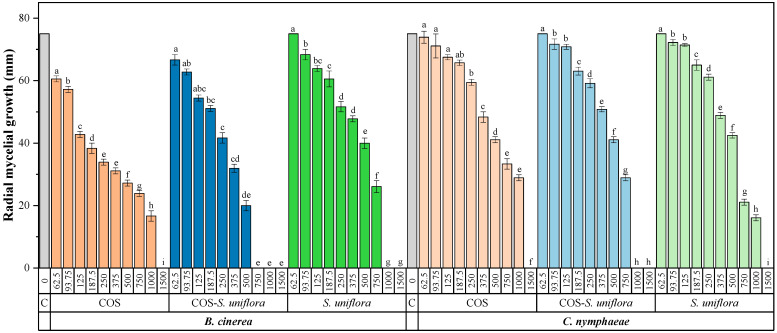
Radial growth values of *B. cinerea* and *C. nymphaeae* in the presence of the different treatments under study—*S. uniflora* extract (*S. uniflora*), chitosan oligomers (COS), and COS-*S. uniflora* extract conjugate complex (COS-*S. uniflora*)—at different concentrations (in μg·mL^−1^). C represents the control. Concentrations of each treatment labeled with the same lowercase letters are not significantly different at *p* < 0.05 according to Tukey’s test. All values are presented as the average of three repetitions, with three replicates per repetition. Error bars represent the standard deviation.

**Figure 4 plants-12-01846-f004:**
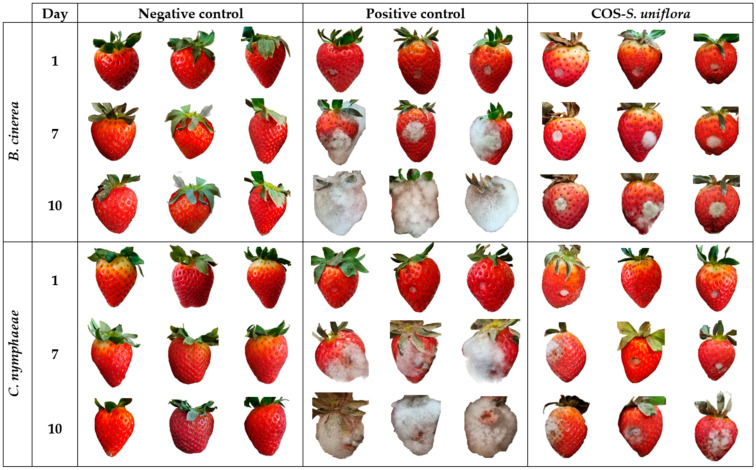
Evolution of the decay of strawberry fruits caused by *B. cinerea* and *C. nymphaeae*: (**left**) negative control, (**center**) positive control, (**right**) treated with COS-*S. uniflora* extract conjugate complex (COS-*S. uniflora*) at MIC×5 (3750 and 5000 μg·mL^−1^ for *B. cinerea* and *C. nymphaeae*, respectively).

**Table 1 plants-12-01846-t001:** Main absorption bands in the infrared spectra of *Silene uniflora* organs, expressed in cm^−1^.

Petals	Fruit	Leaves	Assignment
3366	3335	3366	OH group in phenolic compounds/hydrogen bonding in pyranosides
2955		2954	Symmetric C–H stretching (CH_3_ symmetric stretching)
2916	2918	2915	O–H stretching/C–H stretching
2848		2848	CH_2_ symmetric stretching
1733		1733	C=O stretching, alkyl ester/carboxylic acid (monomeric form)
1706		1706	C=O stretching of carboxylic acid (dimeric form)
1636	1636	1636	Skeletal vibration due to aromatic C=C ring stretching
1472		1472	CH_2_ scissors
1463	1443	1463	Symmetric aromatic ring stretching vibration (C=C ring)
1417	1418	1417	C–H vibration of the methyl group
1378	1371	1378	C–H symmetric bending in CH_3_
1307	1316	1327	CH_2_ wagging, C–O stretching
1243	1244	1243	CH in-plane bending
1146	1146	1147	C–O–C asymmetric stretching
1101	1101		In-plane =C–H bending/C=C stretching
1075			C–O stretching/O–H out plane bending
1018	1019	1020	C–H bending
729		729	CH_2_ rocking
719		719	CH_2_ rocking

**Table 2 plants-12-01846-t002:** Main phytoconstituents identified in the GC-MS chromatogram of *Silene uniflora* extract.

RT (min)	Area (%)	Assignment	Qual
4.6803	0.6939	Pyrrolidine	47
4.7633	0.9593	2-Cyclopenten-1-one, 2-hydroxy-	87
5.7249	0.6416	Succindialdehyde	33
7.3333	0.8584	2(3H)-Furanone, dihydro-4-hydroxy-	38
10.5562	1.1535	2-Methoxy-4-vinylphenol	90
12.3131	0.7600	1-Butanol, 4-butoxy-	32
12.7760	0.9863	3,4-Altrosan	83
16.2164	52.5366	4-*O*-Methyl-*myo*-inositol (mome inositol)	93
17.9160	0.5355	*n*-Hexadecanoic acid	99
25.0918	1.1509	Squalene	99
25.5310	3.6950	Cyclotetracosane	98
26.8367	8.6716	Myristic acid vinyl ester/Palmitic acid vinyl ester	41
26.9614	0.7022	1-Nonadecene/1-heptacosazol	95/93
28.5936	0.9617	Hexadecanoic acid, 4-nitrophenyl ester	51
29.2168	0.6771	1H-Indole, 5-methyl-2-phenyl-	41

Qual: quality of resemblance.

**Table 3 plants-12-01846-t003:** EC_50_ and EC_90_ effective concentrations of *S. uniflora* extract (*S. uniflora*), chitosan oligomers (COS), and COS-*S. uniflora* extract conjugate complex (COS-*S. uniflora*), expressed in µg·mL^−1^, and synergy factors (SF).

Effective Concentration	*B. cinerea*				*C. nymphaeae*			
	COS	*S. uniflora*	*COS*-*S. uniflora*	SF	COS	*S. uniflora*	COS-*S. uniflora*	SF
EC_50_	248	438	236	1.34	674	668	644	1.04
EC_90_	1426	983	746	1.56	721	1420	991	1.46

EC_50_: effective concentration to reduce mycelial growth by 50%. EC_90_: effective concentration to reduce mycelial growth by 90%.

**Table 4 plants-12-01846-t004:** Radial growth of the mycelium of *B. cinerea* and *C. nymphaeae* in the in vitro assays performed on PDA medium loaded with two concentrations (the recommended dose and a tenth of the recommended dose) of three commercial synthetic fungicides.

Commercial Fungicide	Pathogen	Radial Growth of Mycelium (mm)			Inhibition (%)	
		Control	Rd/10	Rd *	Rd/10	Rd *
Azoxystrobin	*B. cinerea*	75	51	12	32	84
	*C. nymphaeae*	75	45	40	40	47
Mancozeb	*B. cinerea*	75	0	0	100	100
	*C. nymphaeae*	75	0	0	100	100
Fosetyl-Al	*B. cinerea*	75	38	0	49.3	100
	*C. nymphaeae*	75	63	0	16	100

* Rd stands for recommended dose, i.e., 62,500 μg·mL^−1^ of azoxystrobin (250,000 μg·mL^−1^ for Ortiva^®^, azoxystrobin 25%), 1500 μg·mL^−1^ of mancozeb (2000 μg·mL^−1^ for Vondozeb^®^, mancozeb 75%), and 2000 μg·mL^−1^ of fosetyl-Al (2500 μg·mL^−1^ for Fosbel^®^, fosetyl-Al 80%). All mycelial growth values (in mm) are average values (*n* = 3).

**Table 5 plants-12-01846-t005:** Disease severity of *B. cinerea* and *C. nymphaeae* attack on cv. “Calinda” strawberries after 1, 7, and 10 days.

Time (Days)	*B. cinerea*			*C.* *nymphaeae*		
	Negative Control	Positive Control	COS-*S. uniflora*	Negative Control	Positive Control	COS-*S. uniflora*
1	0 ± 0 a	0 ± 0 a	0 ± 0 a	0 ± 0 a	0 ± 0 a	0 ± 0 a
7	0 ± 0 a	2.7 ± 0.9 b	0.6 ± 0.2 b	0 ± 0 a	3.3 ± 0.7 b	1.2 ± 0.3 b
10	0 ± 0 a	5 ± 0 c	1.3 ± 0.5 c	0 ± 0 a	5 ± 0 b	2.3 ± 0.8 c

COS-*S. uniflora*: chitosan oligomers-*S. uniflora* extract conjugate complex. Different letters indicate that the disease severity is significantly different at *p* < 0.05.

## Data Availability

The data presented in this study are available on request from the corresponding author.

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
