# Peer review of "Silene uniflora Extracts for Strawberry Postharvest Protection"

_plants, 2023, doi:10.3390/plants12091846_

Round 1

Author Response

The Manuscript titled “Silene uniflora Extracts for Strawberry Postharvest Protection” is written and discussed in depth. The scientific approach is properly designed and developed, and the topic is pertinent to the chosen journal. In this work, the authors studied the bioactive metabolites present in the hydromethanolic extract of Silene uniflora aerial parts through IR and GC-MS analyses. This extract was investigated for the antimicrobial activities, tested in vitro and ex situ, against two strawberry phytopathogens, Botrytis cinerea and Colletotrichum nymphaeae. This activity was improved upon conjugation with chitosan oligomers (COS) and the COS‐S. uniflora conjugate complexes were then tested as protective treatments for postharvest storage of strawberry fruit, obtaining a high protection against the two pathogens.

Some revisions are required:

  • Introduction

Q1. Lines 56: I suggest replacing “by utilizing” with “using”. In the same line after vibrational spectroscopy write (IR), please.

Response: Corrected. The sentence now reads: “[…] (i) to investigate the phytoconstituents of S. uniflora using vibrational spectroscopy (IR) and gas chromatography-mass spectrometry (GC-MS); […]”

  • Results

Q2. Figure 2: There are few errors in a written name. Methyl 2-O-methyl-alpha-D-xylofuranoside: O must be in italic and replace the alpha with the symbol.

Response: Figure 2 has been updated according to the Reviewer’s instructions.

Q3. Paragraph 2.3: Line 93 Eliminate “(Figure)”

Response: It was intended to be a reference to Figure S3 in the supporting information file, but the cross-reference in MS Word was not working. It has been fixed in the revised version of the manuscript.

Q4. Paragraph 2.3: Line 99 Add respectively at the end.

Response: Corrected. The sentence now reads: “[…] reaching full inhibition at 750 and 1,000 μg·mL−1 for B. cinerea and C. nymphaeae, respectively.”

Q5. Figure 3: Lines 102 Please add “(S. uniflora)” after -S. uniflora extract.

Response: Thank you for bringing it to our attention. It has been corrected in the revised version of the text.

Q6. Table 3: Line 113 Please add “(S. uniflora)” after S. uniflora extract.

Response: Corrected, so that all abbreviations used in the table are specified in the table caption.

Q7. Paragraph 2.4: Lines 140 Please write “(positive control)” after treated and add a comma after C. nymphaeae

Response: Corrected. The sentence now reads: “[…] but not treated (positive controls), B. cinerea […]”

  • Discussion

Q8. Paragraph 3.1: Line 173 In the methyl-2-O-methyl-α-D-xylofuranoside, D mustn’t be in italic.

Response: Corrected. Italics has been removed.

Q9. Table 6: Please replace Natural Product with Chitosan complex.

Response: ‘Natural product’ has been replaced with ‘Chitosan complex’ in the table header, following the Reviewer’s recommendation.

  • Materials and Methods

Q9. Paragraph 4.1: Line 295 Please try to explain better the procedure to obtain the samples.

Response: Further details have been included, noting that the specimens were collected during the flowering stage, explaining that they were identified and authenticated by Prof. Dr. Baudilio Herrero Villacorta (Departamento de Ciencias Agroforestales, ETSIIAA, Universidad de Valladolid) and that voucher specimens indeed available at the herbarium of the ETSIIAA.

Q10. Paragraph 4.3: Please add the amount of S. uniflora flowering aerial parts used for the extraction procedure, the volume of the methanol/water solution and the volumes of the solutions used to prepare the COS-S. uniflora complex.

Response: Subsection 4.3 has been updated to include the requested information (“[…] Briefly, 20 g of dried S. uniflora flowering aerial parts were mixed with a 300 mL methanol/water solution (1:1 v/v) and heated in a water bath at 50 °C for 30 minutes. […] COS and S. uniflora extract solutions (150 mL of each solution, both at a concentration of 3000 μg·mL−1) were mixed in a 1:1 (v/v) ratio […]”

  • References

Q11. It is necessary to correct some abbreviations of the names of the journals.

N° 7 Name of journal: Int. J. Pharm. Phytopharmacological Res.

N° 9 Name of journal: Curr. Sci.

N° 19 Name of journal: Phytopathol. Res.

N° 21 Name of journal: Arch. Appl. Sci. Res.

Response: The abbreviations for ‘International Journal of Pharmacognosy and Phytochemical Research’ and the other indicated journals have been corrected.

Reviewer 2 Report

Comments to the Authors-

(1) Abstract-

It is recommended that at the end of this section, the Authors add one sentence to summarize the possible action mode of S. uniflora extracts (and/or COSS. uniflora conjugate complexes) for strawberry postharvest preservation.

(2) Fig.1 &2- Figure legends are too simple. They should be self-explanatory.

(3) L92- 2.3. In vitro Growth Inhibition Tests

Were these in vitro tests for COS, S. uniflora, COS-S. uniflora and three synthetic fungicides performed in parallel simultaneously? What I mean is that comparisons can only be made if the same batch is tested.

(4) L93-  Consider deleting ‘(Figure)’.

(5) Table 5- Consider changing the title from ‘Degree of severity’ to ‘Disease incidence rate’. Alternatively, provide definition and grading criteria of ‘Degree of severity’ in the Materials and Methods section.

(6) L156-160- Parameters such as firmness and color are also very important for evaluating the effect of treatment on fruit quality. Hence, it is suggested the Authors add these data into the manuscript.

(7) L162- 3.1. On the Phytochemical Profile

I recommend that combine all the subparagraphs into one. It is not necessary to separately describe.

(8) L194- 3.2.1. Activity of Other Silene spp. Extracts

I don't see much point or help in just listing these other Silene species for discussion here. Consider removing this paragraph.

Author Response

Comments to the Authors-

Q1. Abstract-It is recommended that at the end of this section, the Authors add one sentence to summarize the possible action mode of S. uniflora extracts (and/or COS‐S. uniflora conjugate complexes) for strawberry postharvest preservation.

Response: We thank the Reviewer for his/her suggestion. Nonetheless, given that we did not investigate the mechanism of action of the extract in detail, but rather provided an educated guess based on previous studies on the mode of action of the three main constituents, we prefer not to include the suggested statement. Please kindly note that, as discussed in subsection 3.2.4, contributions from other minor constituents of the extract and synergistic behaviors among them cannot be ruled out, and the explanation provided for the synergism observed upon conjugation with COS is a hypothesis, so including a statement on the mode of action in the abstract would go against the journal’s guidelines (which establish that “The abstract should be an objective representation of the article: it must not contain results which are not presented and substantiated in the main text and should not exaggerate the main conclusions.”)

Q2. Fig.1 &2- Figure legends are too simple. They should be self-explanatory.

Response: Figure 1 and Figure 2 captions have been updated. The new text is as follows:

“Figure 1. (a) Photograph of Silene uniflora during the flowering stage growing on a cliff in Playa de Cué (Llanes, Asturias, Spain); (b) chemical structure of 2,22-dideoxyecdysone 25-O-β-D-glucopyranoside phytoecdysteroid reported in Silene spp. extracts.”

“Figure 2. Chemical structures of the four most abundant phytochemicals identified by gas chromatrography-mass spectrometry in S. uniflora aerial parts hydromethanolic extract”

Q3. L92- 2.3. In vitro Growth Inhibition Tests. Were these in vitro tests for COS, S. uniflora, COS-S. uniflora and three synthetic fungicides performed in parallel simultaneously? What I mean is that comparisons can only be made if the same batch is tested.

Response: Yes, they were conducted in parallel and using the same source of inoculum. We have updated subsection 4.5.1 to clarify this point: “[…] PDA medium. Tests with commercial fungicides were performed in parallel and using the same source of inoculum. Growth inhibition […]”

Q4. L93-  Consider deleting ‘(Figure)’.

Response: It was intended to be a reference to Figure S3 in the supporting information file, but the cross-reference in MS Word was not working. It has been fixed in the revised version of the manuscript.

Q5. Table 5- Consider changing the title from ‘Degree of severity’ to ‘Disease incidence rate’. Alternatively, provide definition and grading criteria of ‘Degree of severity’ in the Materials and Methods section.

Response: The grading criteria were provided at the end of subsection 4.5.2, but we had not defined it as ‘degree of severity’ but as ‘severity of the disease’, which can be misleading, as noted by the Reviewer. Hence, we have checked the original work by Romanazzi (DOI: 10.1016/j.postharvbio.2012.07.007): the actual term used therein was ‘Disease severity’, so we have updated Table 5 caption, Table 6 header, and subsection 4.5.2 accordingly. We have also noted that it is an empirical scale with six degrees.

Q6. L156-160- Parameters such as firmness and color are also very important for evaluating the effect of treatment on fruit quality. Hence, it is suggested the Authors add these data into the manuscript.

Response: We agree with the Reviewer that information on quality attributes such as firmness and color is relevant. Data on average flesh firmness decreases for the treated samples as compared to the untreated fruits (negative control) has been included in the revised manuscript. Unfortunately, we do not have the necessary equipment (a colorimeter/spectrophotometer) to conduct surface color measurements and provide quantitative values of color changes. We have noted this limitation in the revised version of the article, clarifying that it was a naked-eye qualitative measurement.

The revised paragraph now reads: “Concerning fruit quality attributes, the COS−S. uniflora extract treatment exerted a beneficial effect on the firmness, with an average 24% decrease in flesh firmness values in the case of B. cinerea and a 33% decrease for C. nymphaeae vs. a 52% decrease in the untreated fruits (negative control) by the end of the experiment. As far as color is concerned, the COS−S. uniflora coating imparted a slightly paler shade of red on day 10, more evident in the fruits inoculated with B. cinerea than in those inoculated with C. nymphaeae, although quantitative color measurements would be needed to determine the actual impact on the hue degree and chroma. […]”

Q7. L162- 3.1. On the Phytochemical Profile. I recommend that combine all the subparagraphs into one. It is not necessary to separately describe.

Response: All subparagraphs in subsection 3.1 have been merged into a single one, following the Reviewer’s recommendation.

Q8. L194- 3.2.1. Activity of Other Silene spp. Extracts. I don't see much point or help in just listing these other Silene species for discussion here. Consider removing this paragraph.

Response: The paragraph indicated by the Reviewer has been deleted, as suggested.

Reviewer 3 Report

This paper investigated the antifungi effects of Silene uniflora extracts on strawberry. It’s an interesting topic. However, there are some problems need to be revised.

1.      Introduction need to be rewritten. It’s not concise.

2.      Results part: Vibrational Characterization may be not useful to study. Because the author used the whole part of Silene uniflora as a material to be extracted.

3.      2.2 GC-MS Characterization: In Table 2, for some identified phytochemicals, the Qual was very low. It’s not believable. In this view, the identified phytochemicals are accurate? Subsequently, all discussions about these phytochemicals were questionable.

4.      Figure 3. For C. nymphaeae, radial mycelial growth may be inhibited by S. uniflora at 1250 μg/mL. How did author set the concentration of S. uniflora, COS and the both?

5.      Methods part: all methods were not described clearly. Please improve it.

Author Response

This paper investigated the antifungi effects of Silene uniflora extracts on strawberry. It’s an interesting topic. However, there are some problems need to be revised.

Q1. Introduction need to be rewritten. It’s not concise.

Response: The length of the introduction section has been shortened by almost 30%, from 451 to 327 words.

Q2. Results part: Vibrational Characterization may be not useful to study. Because the author used the whole part of Silene uniflora as a material to be extracted.

Response: Including FTIR data of the dried plant organs used to prepare the extract can validate that the extraction procedure is appropriate, provided that it allows confirming the consistency of functional groups in the starting product with those of the GC-MS-identified phytochemicals in the extract (otherwise the extraction procedure would be leaving valuable phytochemicals behind). FTIR data can also provide additional support for the presence of phytochemicals for which the GC-MS identification is not conclusive (as discussed in the response to Q3 below). Therefore, we deem this vibrational characterization data valuable and opt to retain it.

Q3. 2.2 GC-MS Characterization: In Table 2, for some identified phytochemicals, the Qual was very low. It’s not believable. In this view, the identified phytochemicals are accurate? Subsequently, all discussions about these phytochemicals were questionable.

Response: We are aware of the limitations in the identification of some of the compounds, given that only a small subset of known organic compounds (amenable for GC-MS) is present in the largest mass spectral databases (such as NIST or Wiley). Moreover, the NIST version used by the external laboratory to which the GC-MS analysis was outsourced appears to be an older version (NIST11), which includes a lot less than the later editions, and we do not know what acquisition software was used (e.g., in older versions of Chemstation, matches were low due to the library search comparing the entire mass range of its spectra with the smaller mass range of the acquired data, while Shimadzu's GC-MS solutions do not do this, so matches that would normally record 75-80% register as 95%).

We agree with the Reviewer that reporting compounds with a lower fit brings the interpretation into the grey zone: the identification could have some value, but it could also be completely off. If there was a need for accurate compound identifications for safety assessments, we would certainly avoid reporting compounds with spectral match factors lower than approximately 80%, but -in this case- there are not. Taking into consideration that the occurrence of the main compounds is documented in other plant extracts (subsection 3.1), which would reinforce the database suggestions, and that the functional groups identified in the FTIR spectra of the dried aerial parts data also support the presence of some of the compounds with low ‘Qual’ values (e.g., 2-O-methyl-α-D-xylofuranoside, 2,3-dimethyl-3-hexanol, and 16-hexadecanoyl hydrazide), we have decided to keep the original results section with the ‘best guesses’ obtained from the NIST11 database, but we have included a new paragraph in the discussion to specifically address this Reviewer’s query and make the limitations clear to the readers.

Given that only a small subset of known organic compounds (amenable for GC−MS) is present in the largest mass spectral databases, limitations in the identification of some of the compounds present in the extracts were detected, with quality of resemblance (Qual) values below 80 (Table 2). Caution is advised as identification of such compounds may be unreliable. However, as the main compounds have been documented in other plant extracts (as discussed below) and the functional groups identified in the FTIR spectra of the dried aerial parts data also support the presence of some of the main constituents with low ‘Qual’ values (e.g., 2-O-methyl-α-D-xylofuranoside; 2,3-dimethyl-3-hexanol; and 16-hexadecanoyl hydrazide), the NIST11 database ‘best guesses’ have been deemed acceptable.

Q4. Figure 3. For C. nymphaeae, radial mycelial growth may be inhibited by S. uniflora at 1250 μg/mL. How did author set the concentration of S. uniflora, COS and the both?

Response: The tested concentrations for the extract, COS, and the conjugate complexes are those defined by the European Committee on Antimicrobial Susceptibility Testing (EUCAST), which is among the most popular guidelines used in antifungal susceptibility testing worldwide. They are also the recommended concentrations specified in the Clinical Laboratory Standards Institute (CLSI) standards. In the case of S. uniflora extract against C. nymphaeae, please kindly note that the PROBIT analysis used for the calculation of the 50% and 90% effective concentrations suggests that the EC90 is 1420 μg·mL−1 (see Table 3), which is consistent with the full inhibition found at 1500 μg·mL−1. Consequently, full inhibition at the suggested intermediate concentration (1250 μg·mL−1) is not expected.

Q5. Methods part: all methods were not described clearly. Please improve it.

Response: The Materials and Methods subsection has been improved by addressing several specific issues raised by the other two Reviewers. Subsection 4.1 has been updated, providing details on the phenological stage of the plant when the sampling was collected, and on the identification and authentication of the specimens, apart from information on the moisture content. The amount of S. uniflora dried flowering aerial parts used for the extraction procedure, the volume of the methanol/water solution, and the volumes of the solutions used to prepare the COS-S. uniflora complex have been specified in subsection 4.3. A clarification to point out that tests with commercial fungicides were performed in parallel and using the same source of inoculum has been included in subsection 4.5.1, and details on the equipment used for the firmness measurements have been added to subsection 4.5.2.

Round 2

Reviewer 3 Report

Authors have revised most of problems existed in the manuscript. I still suggest that if you want to identify these important compounds, compound standard can help to identify them using GC. Identification of these compounds is very important to your discussion and conclusion.

Round 3

Reviewer 3 Report

No comments.